# Non-Thermal Plasma—A New Green Priming Agent for Plants?

**DOI:** 10.3390/ijms21249466

**Published:** 2020-12-12

**Authors:** Ľudmila Holubová, Stanislav Kyzek, Ivana Ďurovcová, Jana Fabová, Eva Horváthová, Andrea Ševčovičová, Eliška Gálová

**Affiliations:** 1Department of Genetics, Faculty of Natural Sciences, Comenius University, 842 15 Bratislava, Slovakia; stanislavkyzek@gmail.com (S.K.); ivana.durovcova@gmail.com (I.Ď.); janka.spackova@gmail.com (J.F.); andrea.sevcovicova@uniba.sk (A.Š.); eliska.galova@uniba.sk (E.G.); 2Department of Genetics, Biomedical Research Center, Slovak Academy of Sciences, 845 05 Bratislava, Slovakia; eva.horvathova@savba.sk

**Keywords:** non-thermal plasma, seeds, plants, priming, adaptive response, oxidative stress

## Abstract

Since the earliest agricultural attempts, humankind has been trying to improve crop quality and yields, as well as protect them from adverse conditions. Strategies to meet these goals include breeding, the use of fertilisers, and the genetic manipulation of crops, but also an interesting phenomenon called priming or adaptive response. Priming is based on an application of mild stress to prime a plant for another, mostly stronger stress. There are many priming techniques, such as osmopriming, halopriming, or using physical agents. Non-thermal plasma (NTP) represents a physical agent that contains a mixture of charged, neutral, and radical (mostly reactive oxygen and nitrogen species) particles, and can cause oxidative stress or even the death of cells or organisms upon interaction. However, under certain conditions, NTP can have the opposite effect, which has been previously documented for many plant species. Seed surface sterilization and growth enhancement are the most-reported positive effects of NTP on plants. Moreover, some studies suggest the role of NTP as a promising priming agent. This review deals with the effects of NTP treatment on plants from interaction with seed and cell surface, influence on cellular molecular processes, up to the adaptive response caused by NTP.

## 1. Introduction

It has been more than 150 years since physical plasma was first described and began its journey from research to application in many diverse fields. First used for ozone production and surface treatment of various materials, it gradually penetrated into medicine and life sciences as the development of plasma devices enabled treatment of thermosensitive materials [1]. The discoveries of non-thermal plasma (NTP) effects on living organisms in the field of plasma medicine, such as enhanced wound healing, blood coagulation, or microorganism elimination were very encouraging [2]. Research of NTP´s effects on plant organisms also brought some promising results. Not only can it decontaminate the surface of seeds, but it also enables the enhancement of germination and plant growth. However, the dose of NTP has to be appropriate for a given organism [3,4,5]. Therefore, NTP is not only a positive agent that can be used for growth enhancement but in the first place, it is a stress factor. Organisms, however, have evolved in such a way that the weak stress can cause the reinforcement of an organism, thereby enabling it to tolerate another dose of stress better. This phenomenon is called the adaptive response, priming, or hormesis [6,7]. 

This review aims to describe non-thermal plasma and focus on its effect on plants and the possibility of NTP to induce the adaptive response in plants.

## 2. Characterization of Non-Thermal Plasma

Plasma is one of the basic states of matter. In space, plasma is probably the most widely-occurring state of matter; however, its natural occurrence on Earth is not so common. On the other hand, artificially-made plasma is a part of many devices, such as neon lighting or plasma display panels. In most of these cases, non-thermal plasma is used. NTP is generated by electric discharge in a gas [8] and is usually described as a partially-ionized gas which contains a range of charged and neutral particles. During the generation of plasma, electrons are produced first, the electron density rises to *n_e_* = 10^11^–10^16^ cm^−3^, and they are accelerated in the electric field in a gas medium [9]. Electron density is one of the most important parameters of any plasma [10]. High velocity and density of electrons cause a large number of collisions between the electrons, as well as between the atoms and molecules. As a result, various processes, such as ionization, dissociation, electron attachment, and particle excitation can occur, and products of these reactions eventually become reagents in further reactions. Non-thermal plasma contains not only ions, radicals, and excited molecules but also photons, which are emitted from the dissociation of electrically-excited molecules. The amount of photons is, however, too low to allow photocatalysis to take place [9].

As previously mentioned, there are many chemical reactions in plasma which give rise to a wide range of reactive oxygen and nitrogen species (RONS; O_3_, ^1^O_2_, ∙OH, NO_2_, N_2_O, NO, CO_2_, HNO_3_, HNO_2_, etc.) with various half-lives. The complexity of these chemical reactions arises due to the presence of mono- to multi-atomic molecules in a gas from which plasma is generated [11]. Many plasma devices use neon, argon, or nitrogen as a working gas, in which NTP is generated. Although these gases are inert, an electric discharge gives rise to metastables of Ne, Ar, or N_2_. Metastables have high excitation energy and may play an important role in plasma chemical or ionization processes [10]. Other gases, such as O_2_, air, or the combination of gases are also used for NTP generation. When using air as a working gas, or when a small amount of O_2_ is added to another working gas (usually 0.5–2%), other reactive particles are created, such as ∙OH radical, ozone, or singlet oxygen. Additionally, the addition of small amounts of H_2_O, or water present in a sample treated with NTP increases the production of ∙OH radical, which is very reactive [9,10,11]. NO and other nitrogen oxide species can also be generated in air, or when air is an admixture to the working gas [10].

The final composition of NTP, however, depends on more parameters than just working gas. The type of discharge (dielectric barrier discharge, corona discharge, microwave discharge, gliding arc, plasma jet, etc.), physical input parameters (frequency, voltage, power density, etc.), and gas flow can also influence the NTP composition [9,11,12]. Different configurations of plasma devices require different physical input parameters (amount of energy supplied), which can result in variations in NTP composition or particle concentrations [13]. Therefore, it is challenging to compare the effects of different plasma devices. The most widely-used configurations of plasma devices in biological experiments are dielectric barrier discharge (DBD) and plasma jet configurations (which is also commercially available [14]). DBD usually consists of two electrodes, of which one is connected to a high voltage, while the other is grounded. Either one or both of the electrodes are covered with a dielectric material [15]. Plasma jet is an indirect plasma source, since NTP generated between two electrodes is transported to the treated material using a feed gas (usually helium, argon, or nitrogen). Therefore, the concentration of RONS reaching the treated material tends to be lower compared to DBD, where NTP is in direct contact with the treated material [15]. For an extensive review on plasma reactive species chemistry, mostly in plasma jets, see Lu et al. [10].

Non-thermal plasma can also be referred to as low-temperature or cold plasma, meaning that the temperature of heavy particles (e.g., ions, neutral molecules, radicals) is much lower (~300 K) than the temperature of light particles like electrons (~10^5^ K). Therefore, there is thermodynamic disequilibrium between electrons and heavy particles, and that is why NTP is also called non-equilibrium plasma [11,12]. Another characteristic of NTP is that it is generated either at atmospheric or lower pressure. Non-thermal plasmas used to be generated at a pressure lower than atmospheric due to the easier ionization of gas particles [11]. Thanks to the advances in plasma technologies, NTP can be generated at atmospheric pressure as well [16], which enables the spread of plasma technologies to various fields, such as surface treatment [17], agriculture [18], the food industry [19], degradation of toxic compounds in waters [20,21], and, last but not least, plasma medicine, a field which has been thriving in recent years [22]. 

## 3. Interaction of Non-Thermal Plasma with Biological Material

### 3.1. Effect of NTP on Cells

Non-thermal plasma has been generally used to treat the surface of various materials with the aim of changing the properties of a given surface. Biological material is, however, very complex, and the effect of plasma is not limited only to the surface, but can penetrate much further. Whilst various reactive species in plasma are created, cease to exist, or change in the course of nanoseconds or seconds, in some cases even minutes (H_2_O_2_, NO_2_^−^, NO_3_^−^), their effect on biological material can persist for a longer period of time, i.e. minutes or even days [10]. Typically, more reactive particles have a shorter half-life and can travel a shorter distance before they react (and enable the creation of more stable particles) [23,24,25]. Therefore, reactive particles usually hit only the surface of cells or penetrate only into the surface layer of the cells, and deeper penetration is limited [23]. However, the biological effects of NTP-treatment are not limited only to the surface layer of cells, but also affect cells deeper in animal or plant tissue. This is probably the role of secondary products created from primary RONS of plasma. Secondary products are more stable and have longer half-lives; thus, they can spread the effects of NTP-treatment in space and time [23].

One of the first obstacles on the journey of plasma particles into the cell is the cell wall (in plant and bacterial cells) or an extracellular matrix (in animal cells), where various reactions can occur. Hong et al. [26] simulated the extracellular matrix by 10% foetal bovine serum. They observed that the amount of RONS that got into the synthetic vesicles was reduced up to half when compared to the vesicles in water. RONS can also react with various receptors and signalling molecules in the extracellular space, as well as on the cell surface, triggering signal pathways as effectively as oxidation of molecules inside the cell [26]. Such a receptor might be HPCA1 (hydrogen-peroxide-induced Ca^2+^ increases), leucine-rich-repeat receptor kinase, which is located in the plasma membrane and is activated by extracellular H_2_O_2_. This leads to activation of Ca^2+^ channels and resulting signal transduction [27]. Since RONS originating in NTP are not unfamiliar to living organisms, they can trigger natural defence mechanisms in cells [28].

Eventually, plasma particles can come into contact with a cytoplasmic membrane—a phospholipid bilayer surrounding the cell interior. Application of ROS to a synthetic lipid bilayer resulted in the creation of nanopores [29]. These morphological changes occurred due to the lipid oxidation, and this could be the route for RONS to get from NTP into the cells. Oxidized lipids can also trigger signalling pathways, which can spread the signal, even to cells not directly affected by RONS from plasma [29,30,31]. In addition to the synthetic biological membranes, synthetic vesicles are also suitable for plasma particles interaction studies. Giant unilamellar vesicles (GUV) represent a good model due to their size, which is similar to animal cells, and the possibility to contain molecules such as DNA, proteins, reporter molecules, etc. Ki et al. [31] observed penetration of **·**OH radicals into GUV and also DNA oxidation inside the vesicle. However, the effect inside the GUV was attributed rather to H_2_O_2_ than **·**OH radical, based on the experiments with scavengers of these ROS [31]. The possible interactions of plasma particles with the cell are also summarized in Figure 1. 

The presence of unprotected DNA inside the GUV could also be a model for the interaction of plasma particles with bacteria, which are more sensitive to NTP-treatment than eukaryotic cells. The reason for eukaryotic cells being more resistant to NTP is the DNA protected by a nuclear membrane and a much more effective antioxidant system. This enables the elimination of prokaryotic organisms from the tissue or seed surface without causing damage to eukaryotic cells [32,33]. NTP-treatment can cause physical destruction of prokaryotes or cell death similar to apoptosis [34]. The reason is that RONS from NTP can interact with peptidoglycan in the cell wall and cause its disruption, which can, in turn, result in the physical destruction of the cell and release of its content, or death due to the oxidation stress caused by RONS that got into the cell through cracks in the cell wall. Gram-positive bacteria have a thicker layer of peptidoglycan, and this is probably why they are more resistant to the NTP-treatment than gram-negative bacteria [34,35]. Inactivation of prokaryotic organisms is a field that is well-studied. This impact of NTP-treatment is important, for example, in plasma medicine for surface sterilization of tools or wounds. Elimination of microorganisms using NTP can also be used in the food industry for package and food decontamination, as well as in agriculture for decontamination of seed surface. The use of a DBD plasma device for NTP-treatment of seeds was successful in the elimination of natural microbiota and artificially introduced pathogens, such as *Aspergillus flavus*, *Alternaria alternata*, and *Fusarium culmorum* [36], or different filamentous fungi from the surface of a wheat seed [37]. In addition, Šerá et al. [38] observed the elimination of the phytopathogenic fungi *Fusarium circinatum* from the surface of a pine seed, however, the dose of NTP, which successfully eliminated pathogens, also stopped the germination of seeds.

Plant diseases are usually caused by microorganisms and are very diverse. To fight them, a range of various chemicals is used. However, as the study by Zhang et al. [39] shows, NTP could also be used for the treatment of fungal infections in plants. In their study, NTP was applied to the leaves of *Philodendron erubescens* infected by five fungal pathogens using plasma jet. They managed to reduce the size and number of black spots on leaves or eliminate them altogether. The dose of NTP used was sufficient for inactivation of fungal cells and harmless to the leaves, which is vital for possible future application in the field of agriculture [39]. Such an application of NTP could help lessen the use of pesticides and move us a step closer towards green agriculture. The effect of plasma is not concerned only with surface decontamination, but influences the processes inside the cells of given eukaryotic organisms as well. 

### 3.2. Effect of NTP on Plant Physiological and Biochemical Parameters

In order to use the potential of NTP-treatment, it is inevitable to study the effects of NTP on plants. Usually, NTP is applied to the plant’s seeds, since they are easy to treat, and it is possible to treat the plant in the early stages of development. In most studies, dormant seeds (before imbibition/germination) are treated; however, in some works, germinating seeds (48 h after imbibition) were treated with NTP [40,41]. The advantage of using the treatment of seedling may lie in the fact that many processes are already active and can be more easily affected. There is also more direct contact of NTP particles and the seedling organism, which may result in much shorter treatment times. However, the seedling is not protected by the seed structure and is more vulnerable to plasma constituents (UV, reactive particles, etc.). On the other hand, in the treatment of seeds, plasma can affect the entire embryo, causing faster germination and the formation of a more robust root system, thus increasing the stress resistance of seedling. Moreover, plasma treatment decontaminates the seed surface from pathogenic microorganisms.

In the first studies dedicated to the effect of NTP on plants, enhanced germination and growth parameters of treated plants were mostly observed. Effects of NTP were studied on many plant species–chicory [42], wheat [43,44,45,46], pepper [3,47], maize [36], pea [48,49], hemp [50], cucumber [3], oilseed rape [51], etc. Multiple works stated that there was no change in the final germination percentage, yet the germination speed was enhanced in NTP-treated seedlings [52,53]. On the other hand, there are also studies in which no change in germination or even inhibition of germination was observed [50]. Germination inhibition is usually due to the high dose of NTP. However, an optimal dose of NTP cannot be generally given, since various plant species react differently to NTP-treatment. As an example, Štěpánová et al. [3] determined an optimal treatment time for cucumber seeds at 20 s, but only 4 s for pepper seeds, using their DBD plasma device (power density 100 W/cm^3^). Eventually, a maximal treatment time—a time when a rapid decline in germination was noticed—was also determined (for cucumber–40 s, for pepper—12 s). 

Enhanced germination parameters are related to the enhanced seed wettability or hydrophilicity of the seed surface after NTP-treatment, an effect that many authors have noticed [52,53,54,55,56]. Reactive particles of the plasma attack the seed surface, which results in erosion, etching of surface and/or formation of small cracks. These processes increase surface hydrophilicity [52,54,56] and together with the etched surface of *testa* (seed coat), water can penetrate the seed through the entire surface of *testa*, not only through the standard routes during imbibition. As a result, water uptake to the seed is enhanced. Such a standard route for water uptake is *micropyla*, which is crucial for imbibition in *Phaseolus vulgaris* [52]. However, Alves Junior et al. [53] stated that besides *micropyla*, *hilum* is also essential for water uptake in *Erythrina velutina* after NTP-treatment because hydrophilicity of *hilum* after plasma treatment was enhanced more than that of *micropyla*, thus resulting in more significant water uptake through *hilum* [53]. Etching of the surface or creation of cracks by physical force may not be enough to recreate the enhancement of hydrophilicity by plasma. Gentle scratching of the seed surface also increased hydrophilicity, but not to the same level as NTP-treatment; this is why it is presumed that RONS interactions with surface, which result in micro modifications, are mainly responsible for this effect [56]. As mentioned earlier, plasma particles cannot penetrate very deep into the tissue. After interaction and crossing of external coat layers (*exotesta*), plasma particles probably lose their energy, since inner parts of the seed, like *mesotesta* and cotyledons, already keep their hydrophobicity [52]. Therefore, internal parts of the seed, including the embryo, are protected from the direct impact of plasma particles. However, the size and architecture of the seed tend to be the determining factor of which NTP dose is profitable for given plant species. 

Growth and germination speed are influenced not only by water uptake, but also by activity of enzymes which break-up seed storage material. Sadhu et al. [57] observed an increase in activity of amylases, proteases, and a phytase in NTP-treated seedlings. Dobrynin et al. [58] showed that NTP could change the enzyme activity without seriously damaging the enzyme, probably by the induction of changes in the 3D structure of the protein. In their study, they showed a decrease in trypsin activity after NTP-treatment (4 J·cm^−2^). Changes in lipase activity were also observed by Li et al. [59]. However, they managed to increase its activity using a helium radio-frequency atmospheric pressure glow discharge plasma jet. These authors also attributed the change of enzyme activity to changes in the secondary and tertiary structure of protein due to the presence of plasma reactive particles. 

Švubová et al. [49] focused on the analysis of dehydrogenases in pea seedlings. They discovered that the highest activity of alcohol- and lactate dehydrogenases was in samples with the complete inhibition of germination (180 and 300 s in N_2_-generated NTP). These two enzymes are active mainly during oxygen deficit in the early stages of germination. However, in these samples, the activity of succinate dehydrogenase, which is the enzyme of the Krebs cycle, was much lower. This suggests that in these samples, there was no switch from an-oxygenic to oxygenic metabolism and the embryos suffocated.

The inevitable consequence of the enhanced activity of these enzymes is the increased accumulation of soluble sugars and proteins, as well as overall enhancement in mobilised seed reserve weight, compared to the control sample. It was the increase in the amount of sugars and proteins that many authors observed in particular [39,43,46,57,60]. There was a positive correlation between the quantity of soluble sugars and a plant hormone abscisic acid (ABA) after the NTP-treatment, which suggests that an increment in soluble sugars could be the result of ABA signalling pathway [43]. Zhang et al. [61] also observed enhancement in the amount of soluble sugars and according to them, this could contribute to the boost of germination by improving water retention capacity along with increased water uptake caused by NTP-treatment. Moreover, an increment in proline content was observed as well, which is crucial for water retention in the cell also [43]. In addition, Sadhu et al. [57] noticed the enhanced activity of phytase after the NTP-treatment, which allows for better bioavailability of minerals. Enhanced activity of these enzymes and an increased amount of mobilised seed reserves are the basis not only for the rapid and successful seedling growth, but also for the resistance to adverse conditions [57].

Some studies have also dealt with long-term effects of NTP-treatment, meaning plant growth and fruit production. Shapira et al. [62] did not observe any changes in growth speed, flower and fruit number, nor in shape, colour, and quality of fruits. On the other hand, Yin et al. [63] stated that NTP-treated tomato plants showed better yield, concerning mostly fruit number per plant, but also the weight of individual fruits. Similarly, the better yield was also observed for NTP-treated peanuts. Authors attribute this effect to the larger leaf area and higher chlorophyll and nitrogen contents due to the NTP-treatment of seeds [64]. 

Altogether, the resulting effect of NTP-treatment depends on the source of plasma, parameters of plasma generation (working gas, voltage, power density, frequency), and plant species. The dose of NTP, despite being beneficial for one species, may be phytotoxic for another species, as mentioned previously in the study by Štěpánová [3], which demonstrated species-specific effect.

### 3.3. Changes of Plant Antioxidant Capacity after NTP-Treatment

The action of various stressors on plants usually causes the formation and accumulation of ROS at some point, which is vital for stress signalling. Still, it can also result in oxidative stress. To defend themselves, plants increase the expression or activity of antioxidant enzymes or non-enzymatic antioxidants [65].

Many authors noticed changes in the activity of antioxidant enzymes after NTP-treatment. However, the changes were not consistent, but they were influenced by the type of plasma device, treatment time, working gas, and entry parameters. Zhang et al. [61] observed an increase in the activity of superoxide dismutase (SOD), peroxidase (POD), and catalase (CAT) after the treatment of soybean seeds with argon NTP. A similar observation was made in wheat using DBD air plasma treatment [43]. Helium-plasma treatment of *Catharanthus roseus* seeds also resulted in catalase and peroxidase activities increase [66]. Švubová et al. [49] observed an increase in SOD activity in pea seedlings for all three working gases (air, O_2_, N_2_) using Diffuse Coplanar Surface Barrier Discharge (DCSBD; volume power density 80 W/cm^3^), except for the treatment times that completely inhibited the germination (180 and 300 s in N_2_-generated NTP).

On the other hand, some works did not provide such clear results. In the study by Henselová et al. [67], which describes the effects of NTP-treatment on antioxidant enzymes activities of maize seeds, only small, usually non-significant changes, compared to the control samples, were found. Tong et al. [68] noticed an increase only in the activity of CAT, and only for one experimental group (5950 V, 10 s) of *Andrographis paniculata* plants. Moreover, with regard to SOD, they observed a decrement in its activity in several groups (lower voltages during NTP generation: 3400, 5100, 4250 V) and no change in POD activity. Despite the minimal changes in CAT activity, the authors noticed that CAT isoenzyme expression showed significant changes in almost every group [68]. The importance of working gas for differences in the activity of antioxidant enzymes is illustrated in the study by Kabir et al. [45]. They observed changes in antioxidant enzyme activity between NTP generated in Ar/O_2_ and Ar/air mixtures. NTP generated in Ar/O_2_ induced an increment in SOD and ascorbate peroxidase (APX) activity, while no changes were found for Ar/air plasma. The pre-treatment of wheat seeds by Ar/O_2_ plasma could even reverse the decrease in antioxidant enzymes activity caused by cadmium [45].

Plasma treatment of wheat seedlings resulted in increased activity not only of peroxidase, but also of phenylalanine ammonia lyase (PAL) [40,44,47], which is a key enzyme of polyphenol compounds biosynthesis that also participates in plant protection from various stresses; its activity is usually induced by some stresses or hormones [69]. The activities of both enzymes were increased even in the seedlings pre-treated with NTP and subsequently exposed to salinity stress [44], a high concentration of selenium nanoparticles [40], or ZnO nanoparticles [47] compared to the samples not pre-treated with NTP. Such an increase in antioxidant capacity of NTP-pre-treated seedlings likely contributed to the higher stress tolerance. 

Depending on the dose of NTP, the type of plasma device and antioxidant capacity of plant cells, NTP-treatment of seeds or plants can also cause the oxidation of macromolecules, since NTP contains RONS. Such macromolecules are, for example, lipids. Malondialdehyde (MDA) is widely used as a marker of lipid peroxidation because it is a product of polyunsaturated fatty acid peroxidation [70]. Zhang et al. [61] stated that the application of argon plasma on soybean seeds reduced MDA concentration compared to the control sample. At the same time, increased activity of antioxidant enzymes was observed. Taken together, this suggests increased stress resistance, which can also contribute to the enhancement of germination. Additionally, Tong et al. [68] were able to set parameters of plasma treatment so that the MDA concentration decreased, and, at the same time, the activity of catalase increased. This negative correlation was also observed after DBD plasma treatment of wheat seeds [43]. MDA can also be a part of the upregulation of some genes involved in abiotic stress response [71], and this supports the idea of non-thermal plasma as an inductor of adaptive response (priming agent).

Even though plant secondary metabolites are not essential for the functioning of the plant, they are still very important. They represent a considerable pool of molecules that facilitate the everyday life of a plant and help it to adapt to its environment and survive. Various plant secondary metabolites are essential for reaction to different stressors, especially phenolic compounds, which are a part of antioxidant activities and enhanced tolerance to stress. As previously mentioned, NTP-treatment can enhance the activity of PAL, which is a key enzyme of polyphenols biosynthesis. The treatment of seeds of *Echinacea purpurea* with NTP resulted in an increased amount of some compounds, for example, vitamin C or phenolics. Radical scavenging activity also increased in proportion to the amount of phenolic compounds [72]. 

### 3.4. Changes in Plant Gene Expression after the NTP-Treatment

One of the natural responses to stress is manifested in gene expression changes. Organisms need gene products, which can help them overcome adverse conditions. This is particular in the case of plants, since they are sessile and thus unable to escape unfavourable conditions. This is reflected, for example, in the diversity of heat shock proteins in plants [73]. 

Plant heat shock factors (HSF) are essential transcription factors that regulate several genes that affect the growth and development of a plant, but also genes that are a part of a stress response [74]. HSFA4A is a substrate for the MPK3/MPK6 signalling pathway and is a part of a response to stress conditions in *Arabidopsis thaliana* [75]. An increase in the expression of this HSF was observed in roots and shoots of NTP-treated wheat seedlings, with the induction of expression being faster in roots than shoots [44]. Zhang et al. [61] focused on the effect of the NTP-treatment on the expression of selected genes. They discovered that some genes for ATP synthesis, *GRF1-6 (growth-regulating factor)* and *TOR (target of rapamycin)* genes had an increased expression, which correlated with a decrease in methylation of given loci. Moreover, functions of these genes are closely related to metabolism and growth regulation, and the authors therefore point out that reduced methylation (and thus increased expression of the mentioned genes) may have contributed to the improvement of germination and growth. 

Plasma-treatment of seeds also affects the expression of drought-responsive genes. After NTP-treatment of wheat seeds, a decrease of *LEA1* gene expression was observed and conversely, there was increased expression of *SnRK2* and *P5CS* genes [43]. The product of the *SnRK2* gene is one of serine/threonine kinases that is specific for plants and is a part of plant response to abiotic stress. *P5CS* is one of the key genes of proline biosynthesis. Moreover, in the study by Guo et al. [43], a positive correlation between the increase in proline and ABA content, as well as *SnRK2* and *P5CS* expression was observed, which might indicate that *SnRK2* is a part of ABA signalling during abiotic stress, since some *SnRK2* (not all of them) are ABA-responsive [76].

NTP also altered the expression of genes for antioxidant enzymes. The application of Ar/O_2_ plasma alone resulted only in an increase of *TaSOD* gene expression. Still, the combination of NTP-pre-treatment with later exposure of seedlings to cadmium caused higher expression not only of *TaSOD,* but also *TaCAT*, compared to the positive control (cadmium only). In addition to genes for antioxidant enzymes, two genes encoding cadmium transporters (*TaLCT1* and *TaHMA2*) have also been investigated. For both genes, cadmium caused their overexpression; however, if the seedlings were pre-treated by NTP, their expression was at the level of negative control, despite the presence of cadmium [45].

### 3.5. Genotoxic Effects of NTP in Plants

Since NTP can emit a certain amount of UV radiation, Shapira et al. [62] focused on the detection of pyrimidine dimers (CPD; cyclobutane-type pyrimidine dimers and the pyrimidine (6,4) pyrimidone dimers) in the DNA of cotyledons of tomato and pepper seeds, but did not detect an increased amount for the whole seed treatment or the direct treatment of cotyledons. For a comparison, the direct irradiation of cotyledons (without *testa*) with UV caused a significant amount of dimers, but when whole seeds were treated (with *testa*), the number of dimers was minimal. This suggests that *testa* can protect the seed and the embryo from UV radiation [62]. Kyzek et al. [48] also focused on the detection of DNA damage caused by NTP-treatment (DCSBD, generated in the air) of seeds using alkaline comet assay. The amount of DNA damage in pea seedlings pre-treated with NTP did increase slightly with increasing treatment time (60–300 s), yet there was no significant difference compared to the negative control (without plasma pre-treatment) [48]. NTP generated in ambient air indeed seems to be the most suitable treatment for pea seeds. Tomeková et al. [77] compared the DNA damage caused by NTP generated in ambient air, oxygen, nitrogen, or mixtures of these two gases. The DNA damage was the lowest in seedlings treated with ambient air NTP. On the other hand, NTP generated in pure nitrogen caused the highest amount of DNA damage; however, with the increasing concentration of oxygen and decreasing concentration of nitrogen in working gas, the DNA damage decreased, but the amount was still higher compared to the ambient air NTP. To date, there are few works that have studied the effect of NTP-treatment on DNA, but so far it seems that treatment times, which have an enhancing impact on germination, growth, and tolerance to stress, are genotoxicologically safe. 

## 4. Non-Thermal Plasma as an Inductor of Adaptive Response to Abiotic Stress

In nature, the ideal conditions for organisms, including plants, occur relatively rarely, compared to those which are unfavourable. Therefore, plants, as well as other living organisms, have evolved mechanisms or strategies that enable them to actively defend themselves from a variety of adverse conditions, allowing them to at least tolerate any unfavourable conditions [78]. Stress factors can be adverse abiotic conditions (drought, salinity, high or low temperatures, etc.), diseases caused by viruses, bacteria and other parasites, herbivores, which are usually much bigger than the plant, or even other plants in the area that fight for space, nutrients, and water. All of these agents can act with varying intensity, either alone or (in most cases) in combination with other stressors [79]. 

The intense action of the stress or stressors is a burden on the plant. Therefore, it must redirect the energy intended for growth and reproduction to stress signalling and defence mechanisms, which significantly reduce the fitness of such plant. Although the low intensity of stress can “force” the plant to trigger defence mechanisms, it is not too large a burden for a healthy plant, and therefore, the decrease in fitness may not be significant [80]. Moreover, it can have a positive effect in the long run. Such a mechanism denoted as priming, hormesis, or adaptive response may, in some way, resemble vaccination in humans and animals. These terms represent processes, in which a mild dose of stressors can induce signalling pathways that “prepare” or prime the organism for the possible action of another stressor. The response to another, stronger stress is usually quicker and more aggressive than in organisms that were not primed [81]. However, the initial stress must not be too strong; otherwise it can significantly weaken the organism, or even kill it [82].

As previously mentioned, the action of primary stress (stimulus) elicits a specific defensive reaction in the plant. For a plant to respond quicker and more effectively to the effects of further stress, it must have some stored information about the initial stimulus and how to respond to it. This could be the accumulation of specific molecules (e.g., dormant transcription factors, signalling molecules), protein modification, changes in the epigenome, etc. [83]. It is not yet fully elucidated for how long the primed plant can store such information. Still, it may depend on the sustainability of these changes and the intensity of the initial stimulus [84]. It has been observed that the information about the heat stress priming of parental and F_1_ generation was transferred even to the F_3_ generation, which subsequently tolerated the effects of heat stress better, as well as had a better fitness (in terms of flower number and seeds produced) compared to the F_3_ generation of the line, in which parental and F_1_ generation did not undergo priming. In this case, it is probable that the information about stress was stored in the form of epigenetic changes, since the authors of the study, on the basis of experimental design, did not consider mutations or actions of other genetic factors to be the cause of this transgenerational transfer of stress information [85].

In several studies, NTP has been used as the elicitor of the adaptive response. Various effects of NTP-treatment on plants were observed, and some of them are illustrated in Figure 2. Pre-treatment of seedlings with NTP has been shown to be a suitable priming agent for the toxicity of the higher doses of selenium (nSe) and zinc oxide nanoparticles (nZnO). In this study [40], seedlings (48 h after imbibition) of *Melissa officinalis* were exposed to NTP-treatment. Those treated only with NTP showed positive effects on growth parameters (root and shoot length, leaf area, fresh weight). The negative impact of high concentrations of nanoparticles on seedlings growth was milder in seedlings pre-treated with NTP. For example, nSe concentration of 10 mg/L caused severe growth inhibition, however, the seedlings pre-treated with NTP and subsequently exposed to the same concentration of nSe showed a significant improvement in all growth parameters [40]. Additionally, in *Cichorium intybus*, the NTP-treatment mitigated the phytotoxic effect of a high dose of nSe. The authors observed an increase in catalase and peroxidase activity as well, which probably alleviated the oxidative stress caused by nanoparticles. In long-term observations, the authors found that plants pre-treated with NTP showed better growth parameters, such as shoot fresh weight, root biomass, flower number, and flower fresh weight. It is the reinforcement of the root system in NTP-treated plants that may have been an important aspect of better growth, since the root system plays a crucial role during seedling growth [42]. Faster root growth was also observed in the NTP-pre-treated pepper (*Capsicum annuum cayenne*) despite ZnO-nanoparticles-treatment, compared to the ZnO-treatment only. There was also an improvement in parameters important for tolerance or stress management, such as the increased activity of peroxidase, PAL, an increased amount of carotenoids and chlorophyll a, and soluble phenols [47]. Therefore, in this case as well, the NTP-treatment was able to mitigate the deterioration of growth parameters caused by ZnO nanoparticles. Moreover, by the addition of plant hormones to the growth medium, the effect of plasma treatment was amplified, thereby suggesting that NTP-treatment can alter cellular sensitivity to the hormones, and that the composition of the culture medium may modify the plasma–plant interactions [47]. 

The impact of heavy metals on plants is well-known, and is a serious environmental problem, especially in mining areas. In the study by Kabir et al. [45], cadmium caused deterioration in the growth parameters of wheat, a decrease in the amount of chlorophyll and soluble proteins, and, on the contrary, an increase in the amount of cadmium in the plant, as well as the amount of H_2_O_2_, expression of *TaLCT1*, and *TaHMA2* genes encoding cadmium transporters. Pre-treatment of seeds with NTP managed to ameliorate all these symptoms and also caused an increase in CAT and SOD activities, as well as an increase in expression of their genes. In addition, decreased expression of *TaLCT1* and *TaHMA2* corresponded with decreased concentration of cadmium in the plant. The reduction of its concentration could also be due to the decreased seed pH or growth medium pH because of NTP-treatment, since the bioavailability of cadmium can be reduced at low pH. 

Cells have to cope with DNA damage every day. There are intracellular processes that can cause damage (transcription or ROS production), but there are also damaging agents, such as UV light, which can be present in the environment [86]. Kyzek et al. [48], in their study, used zeocin, a radiomimetic antibiotic, to induce DNA breaks. They discovered that pre-treatment of pea seeds with NTP resulted in a reduction of DNA damage caused by zeocin up to half. The decrease of damage, however, was dependent on the treatment time of NTP (60–300 s), with the most substantial priming effect for 180 and 240 s treatments. 

Mitigation of the effects of drought stress was also observed in seedlings pre-treated with NTP. Plasma pre-treated seedlings not only had better growth in drought conditions, but also increased amounts of proline, and soluble sugars, as well as increased expression of genes characteristic for plant stress response, such as *SnRK2* and *P5CS*. Reduced quantities of MDA and ROS, decreased expression of *LEA1* gene, and an increase in antioxidant enzymes activity and ABA hormone amount were also observed, when compared to the non-primed plants [43]. Therefore, in addition to drought-specific reactions, more general responses, such as increased antioxidant activity and plant hormone regulation also contribute to the increased stress tolerance. Similarly, NTP priming of tomato alleviated the effect of drought stress mediated by polyethylene glycol (PEG) suggesting the role of H_2_O_2_ and NO signalling in plant defence [87]. The effect of H_2_O_2_ and NO as signalling molecules in barley seeds pre-treated with NTP was also discussed by Gierczik et al. [88]. In Arabidopsis, the positive effect of NTP pre-treatment was lower under osmotic stress induced by PEG than under saline stress [89].

In the study by Seddighinia et al. [41], it was shown that NTP-pre-treatment is not only able to mitigate the adverse effects of stress, but also to amplify its positive impact, if the primed plant is subsequently exposed to another priming agent. In this case, the second priming agents were multi-walled carbon nanotubes. Plasma caused an increased intake of nanotubes into the plant. Together, these priming agents managed to improve several growth parameters better than NTP or nanotubes alone. 

The results of all these studies suggest that NTP could be used as a novel priming agent for a variety of adverse conditions that occur naturally in a plant environment, since the doses of NTP which are safe for plants can usually enhance growth. Still, most importantly, they can induce the adaptive response that results in the mitigation of adverse effects of stresses. 

## 5. Conclusions

In this review, we summarized the effects of NTP-treatment on plants. When the treatment conditions are optimized, NTP may enhance the growth parameters and affect processes that are a part of a reaction to stress. Indeed, NTP acts as a stressor, since it is composed mainly of RONS and charged and neutral particles; however, it is beneficial when we aim to induce an adaptive response to enhance the tolerance or survival of organism during subsequent stress(es). Non-thermal plasma may represent a promising novel priming agent, not only because of the induction of the adaptive response, but also because of its ability to sterilize the seed surface and therefore prevent growth retardation or various diseases caused by pathogens. 

Seeds treated with NTP might represent an environmentally-friendly approach to producing seeds with the potential of growing into a plant that is more resistant to stresses occurring during germination and early growth. As a consequence, this may result in a plant being able to produce more biomass or fruits, even under challenging environments, which might be very useful considering the serious changes to the world´s climate. However, such environments are not exclusively terrestrial. Growing plants in space stations or on bases on the Moon or Mars has its own difficulties, which might be, at least partially, alleviated by NTP treatment of plant seeds. 

## Figures and Tables

**Figure 1 ijms-21-09466-f001:**
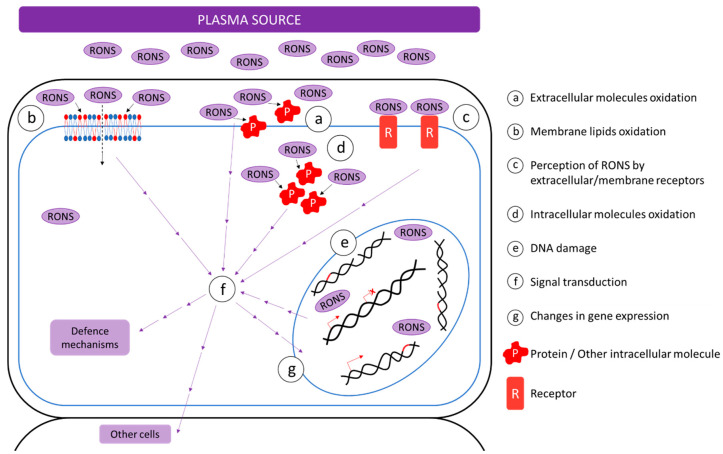
Illustration of possible interactions of non-thermal plasma (NTP) particles with cells. Reactive particles from plasma can oxidize molecules in the extracellular space or lipids in the cytoplasmic bilayer, as well. The latter could result in the creation of nanopores—a possible route for RONS to get inside the cell and oxidize molecules in the cytoplasm or get into the nucleus and cause DNA damage. Particles outside the cell can also interact with appropriate receptors. Oxidized molecules or membrane lipids, activated receptors, and damaged DNA can all transduce a signal about the increased amount of RONS, and the cell can react by changes in gene expression, activation of defence mechanisms, or signal transduction to other cells.

**Figure 2 ijms-21-09466-f002:**
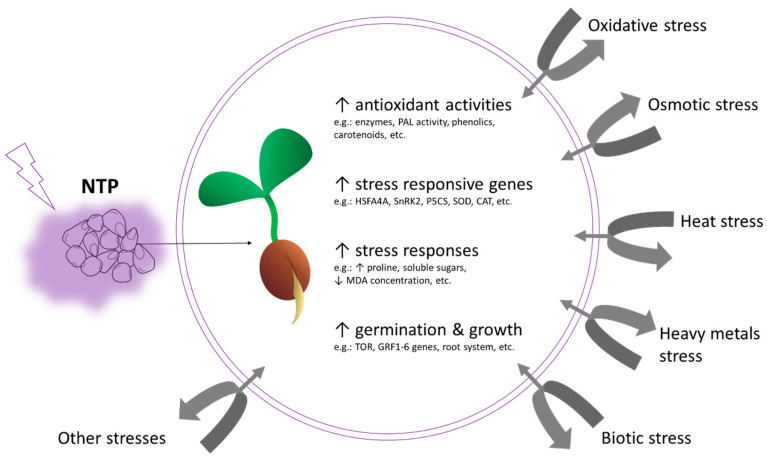
Illustration of some effects of non-thermal plasma on plants. Usually, seeds are treated with NTP. Eventually, the seedling may have an increased activity of antioxidant enzymes or non-enzymatic antioxidants, expression of various genes, usually related to stress response (e.g., heat shock factor *HSFA4A* or *SnRK2*) or metabolism (e.g., *TOR, GRF1-6*), or increased amounts of e.g., proline or soluble sugars. All these reactions may help a plant tolerate more severe stresses that may occur in the future.

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
