# Peer review of "Non-Thermal Plasma—A New Green Priming Agent for Plants?"

_ijms, 2020, doi:10.3390/ijms21249466_

Round 1
Reviewer 1 Report
I believe that the manuscript is outstanding and can be published after some minor linguistic check.
I would add something is more information regarding the phytotoxicity. And maybe to clear up concentration issues regarding the levels by which it is considered beneficial and the levels it is considered phytotoxic.
Author Response
Response to Reviewer 1 comments
We thank the Reviewer 1 for careful reading of the manuscript and providing valuable comments and suggestions that increase the quality of the manuscript.
Point 1: I believe that the manuscript is outstanding and can be published after some minor linguistic check.
Response 1: As you suggested, we have had the manuscript checked by the native speaker and we corrected the mistakes.
Point 2: I would add something is more information regarding the phytotoxicity. And maybe to clear up concentration issues regarding the levels by which it is considered beneficial and the levels it is considered phytotoxic.
Response 2: Thank you for this comment regarding phytotoxicity. We added a short paragraph about the plasma phytotoxicity (lines 256-259). The concentration/dose of plasma which might be phytotoxic depends on the type of plasma source, working gas, plasma generation parameters and length of plasma treatment, as we tried to illustrate in chapter 2. Moreover, plasma treatment of seeds is species specific (as shown in Štěpánová et al., 2018).
Reviewer 2 Report
In this review, authors summarized and discussed the research related to plasma effect on plants and priming plants. Authors’ discussion is quite systematic, and plasma priming is a timely subject for discussion. Here are comments.
- Titles in section 3 need to be more organized. 3.2 – 3.5 are mostly related to plants. 3.2 title is too general. Either modify title of section 3 more specific or add word “plant” in titles of section 3.4 and 3.5?
- Section 4 addressed plasma priming for mostly abiotic stresses. What about biotic stresses? If abiotic stresses are focused, better modify section title. In addition, several papers related to abiotic stress responses are missing.
Bafoil, M.; Le Ru, A.; Merbahi, N.; Eichwald, O.; Dunand, C.; Yousfi, M. New insights of low-temperature plasma effects on germination of three genotypes of Arabidopsis thaliana seeds under osmotic and saline stresses. Sci. Rep. 2019, 9, 8649, doi:10.1038/s41598-019-44927-4.
Gierczik, K.; Vukušić, T.; Kovács, L.; Székely, A.; Szalai, G.; Milošević, S.; Kocsy, G.; Kutasi, K.; Galiba, G. Plasma-activated water to improve the stress tolerance of barley. Plasma Process. Polym. 2020, 17, e1900123, doi:10.1002/ppap.201900123.
Adhikari, B.; Adhikari, M.; Ghimire, B.; Chandra Adhikari, B.; Park, G.; Choi, E.H. Cold plasma seed priming modulates growth, redox homeostasis and stress response by inducing reactive species in tomato (Solanum lycopersicum). Free Radic. Biol. Med. 2020b, 156, 57-69, doi:10.1016/j.freeradbiomed.2020.06.003.
- Several sentences are not good in readability.
For example, lines 16 -17, 19-20, 98-99, 182-187, 383-385
- Line 115-117, is this sentence correct?
“Athough only a small amount of RONS get into cells, we cannot exclude that they will not react with various receptors and signalling molecules outside the cell - in the extracellular space and on the cell surface.”
Author Response
Response to Reviewer 2 comments
We thank the Reviewer 2 for careful reading of the manuscript and providing valuable comments and suggestions that increase the quality of the manuscript.
Point 1: Titles in section 3 need to be more organized. 3.2 – 3.5 are mostly related to plants. 3.2 title is too general. Either modify title of section 3 more specific or add word “plant” in titles of section 3.4 and 3.5?
Response 1: Considering your suggestion, we would rather not modify the title of section 3 as it deals with plasma interaction with various biological materials including artificial cells and microorganisms, not only plants. However, we modified the titles in section 3.4 and 3.5 by adding the word “plant”, according to your suggestion. These changes better reflect the content of sections 3.4 and 3.5. We also modified title in section 3.2 to be more specific.
Point 2: Section 4 addressed plasma priming for mostly abiotic stresses. What about biotic stresses? If abiotic stresses are focused, better modify section title. In addition, several papers related to abiotic stress responses are missing.
Bafoil, M.; Le Ru, A.; Merbahi, N.; Eichwald, O.; Dunand, C.; Yousfi, M. New insights of low-temperature plasma effects on germination of three genotypes of Arabidopsis thaliana seeds under osmotic and saline stresses. Sci. Rep. 2019, 9, 8649, doi:10.1038/s41598-019-44927-4.
Gierczik, K.; Vukušić, T.; Kovács, L.; Székely, A.; Szalai, G.; Milošević, S.; Kocsy, G.; Kutasi, K.; Galiba, G. Plasma-activated water to improve the stress tolerance of barley. Plasma Process. Polym. 2020, 17, e1900123, doi:10.1002/ppap.201900123.
Adhikari, B.; Adhikari, M.; Ghimire, B.; Chandra Adhikari, B.; Park, G.; Choi, E.H. Cold plasma seed priming modulates growth, redox homeostasis and stress response by inducing reactive species in tomato (Solanum lycopersicum). Free Radic. Biol. Med. 2020b, 156, 57-69, doi:10.1016/j.freeradbiomed.2020.06.003.
Response 2: It is right, we focused on abiotic stress, therefore we modified the section 4 title by adding abiotic stress in it. We also substituted “priming agent” with “inductor of adaptive response” for better readability of the title.
We included suggested papers into section 4, lines 463-468.
Point 3: Several sentences are not good in readability. For example, lines 16 -17, 19-20, 98-99, 182-187, 383-385
Response 3: Suggested sentences were re-written. The change in line 179 was made due to changes in lines 182-187.
Point 4: Line 115-117, is this sentence correct?
“Although only a small amount of RONS get into cells, we cannot exclude that they will not react with various receptors and signalling molecules outside the cell - in the extracellular space and on the cell surface.”
Response 4: Sentence in lines 115-117 (now 116-118) was also re-written and, hopefully, simplified.